# Associations between Vaccination Behavior and Trust in Information Sources Regarding COVID-19 Vaccines under Emergency Approval in Japan: A Cross-Sectional Study

**DOI:** 10.3390/vaccines11020233

**Published:** 2023-01-20

**Authors:** Hiroko Okada, Tsuyoshi Okuhara, Eiko Goto, Takahiro Kiuchi

**Affiliations:** Department of Health Communication, School of Public Health, The University of Tokyo, Tokyo 113-8654, Japan

**Keywords:** COVID-19, pandemic, vaccination, trust, information, health communication

## Abstract

We examined the association between COVID-19 vaccination behavior and trust in COVID-19-related information sources during the initial period of COVID-19 vaccination in Japan. A cross-sectional survey was conducted in August 2021, 5 months after the start of COVID-19 vaccination for the general public under emergency approval. Participants were recruited using non-probability quota sampling from among Japanese residents who were under a declared state of emergency. Sociodemographic data, vaccination behavior, and levels of trust in eight media sources of information and three interpersonal information sources were assessed using an online survey form. A total of 784 participants completed the survey. The results of multiple logistic regression analysis showed that age, household income, underlying medical conditions, and living with family were significantly associated with COVID-19 vaccination behavior. Regarding COVID-19 vaccine information sources, trust in public health experts as a source of media information and primary care physicians as a source of interpersonal information showed significantly positive associations with COVID-19 vaccination behavior (odds ratio [OR] = 1.157, 95% confidence interval [CI] 1.017–1.31; OR = 1.076; 95% CI 1.006–1.150, respectively). Increasing trust in public health experts and primary care physicians and disseminating vaccine information from these sources will help promote vaccination under emergency approval.

## 1. Introduction

Coronavirus disease 2019 (COVID-19) has spread rapidly across the world, with serious impacts on human health and society. With effective treatment for COVID-19 infection lacking, vaccination is the cornerstone of infection control. COVID-19 vaccines were developed rapidly, and by the end of 2020, the BNT162b2 (Pfizer–BioNTech) vaccine, mRNA-1273 (Moderna) vaccine, and Ad26.COV2.S (Johnson & Johnson–Janssen) vaccine had received Emergency Use Authorization (EUA) from the U.S. Food and Drug Administration [1,2]. The same vaccines that obtained EUA in the United States (U.S.) also received emergency approval in Japan in February 2021. People had to accept a greater degree of uncertainty and overcome greater anxiety than with regular vaccines if they chose to receive the COVID-19 vaccines, which were developed in less than a year and offered under emergency approval [3].

A key factor influencing vaccination behavior is trust [4,5]. Trust is based on the relationship between two actors: the trustor (the one who places their trust in another) and the trustee (the one who is trusted) [6]. There are various definitions of trust, but in general, trust is a voluntary expectation on the side of the trustor that the interaction with the trustee will lead to gains rather than losses [6]. Cooperation is a consequence of trust, and mistrust is likely to decrease cooperation [7].

During the pandemic, people were exposed to various sources of information, such as the government and experts [8]. Many studies have reported that trust in government is a factor that promotes public acceptance of and cooperation with government-recommended public health practices. Regarding vaccination, people’s trust in their government is associated with the acceptance of vaccinations against H1N1 influenza (i.e., swine flu) [9], as well as perceptions of seasonal influenza vaccination [10] and COVID-19 vaccination intentions [11,12,13]. However, most past studies have examined cognitive factors as outcomes, such as behavioral intention and acceptance, rather than vaccination behavior. In a study during the early stages of the COVID-19 pandemic, the U.S. Centers for Disease Control and Prevention (CDC) was one of the most frequently cited sources of trusted information among U.S. residents [14]. In Japan, the Novel Coronavirus Expert Meeting (later known as the Advisory Board for Countermeasures against Novel Coronavirus Infections), comprising experts in infectious disease control, was organized in February 2020 to provide advice from a medical standpoint on countermeasures against COVID-19. However, it is unclear whether trust in this temporary expert group was at the same level as the level of trust in the U.S. CDC among Americans. In an information-driven society, any individual (such as a patient or celebrity) can disseminate messages to large numbers of people through various media, including the Internet and social media [15]. With COVID-19, there has been large-scale exposure to information from different sources [16,17]. Thus, health information that individuals trust during a pandemic and that leads to vaccination behavior is not limited to government sources.

The sources of information in which people placed their trust during the COVID-19 pandemic were not limited to media sources. Trust in information sources, such as information received in face-to-face interpersonal communication, has also been reported to be associated with vaccination [18,19,20,21,22,23,24,25,26,27,28]. During the early stages of the H1N1 influenza pandemic, the information source most trusted by the U.S. public was their primary care physician [18]. An association between distrust of physicians and hesitancy to receive the H1N1 influenza vaccine has also been reported [19,20,21]. Furthermore, physicians’ recommendations and communication with physicians are key promoters of vaccination [22,23,24,25,26,27,28]. Trust in physicians as a source of information may also be a factor that promotes COVID-19 vaccination.

As mentioned, most previous studies examining the association between trust in information sources and vaccination have focused on trust in governments, and findings on the association between COVID-19 vaccination behavior and trust in physicians, experts, and other individuals are limited. Additionally, most outcomes in past studies were cognitive factors rather than behavior. The purpose of this study was to fill gaps in these findings by examining the association between trust in multiple sources of information and vaccination behavior. Our results will inform communication strategies when initiating new vaccination campaigns during an emerging infectious disease pandemic, such as disseminating information from highly trusted sources, which has a powerful influence on people’s vaccination behavior.

## 2. Materials and Methods

### 2.1. Study Design

This was a cross-sectional study in Japan. Web-based surveys were conducted in August 2021. In Japan, the COVID-19 vaccine received emergency approval in February 2020, and the vaccination of healthcare providers began. For the general public, vaccination of people 65 years and older started on 12 April 2021, and vaccination of people younger than 65 years started on 1 June 2021. Surveys were conducted 5 months after the start of vaccination for the general public aged 65 years and older and 2.5 months after the start of vaccination for the general public aged less than 65 years. The emergency approval system for drugs in Japan allows the use of drugs distributed in other countries without the need for domestic clinical trials. At the time of the survey, the legal status of the COVID-19 vaccine in Japan was “Duty to Endeavor” for people aged 12 and older. This means that vaccination was not mandatory, but efforts had to be made to vaccinate.

### 2.2. Data Collection

The sample was recruited using the platform of a research company, Rakuten Insight, Inc., from a panel of 2.2 million Japanese residents registered in the company’s database. Inclusion criteria for the study were as follows: (1) residents of areas where a state of emergency had been declared as of August 2021; and (2) aged 18 years or older. Exclusion criteria were (1) unable to be vaccinated owing to a physical condition and (2) being a health care provider. Emails were sent to 8774 survey company enrollees who met the eligibility criteria, excluding healthcare providers. Of those, 1565 e-mail recipients accessed the text detailing the survey and indicated their willingness to participate. Of those, age, gender, and prefecture of residence were sampled to match Japan’s general population, and 788 were invited to participate in the web survey. Four respondents who reported that they were unable to be vaccinated due to physical conditions were excluded. Seven hundred eighty-four were included in the final analysis. The survey was conducted on 15–16 August 2021. A total of 784 people from eight prefectures (Chiba, Hokkaido, Hyogo, Kanagawa, Kyoto, Osaka, Saitama, and Tokyo) that were under a state of emergency owing to the fifth epidemic wave of COVID-19 completed the survey.

### 2.3. Measurements

Sociodemographic data included sex, age, education, household income, place of residence, employment status, presence of underlying medical conditions, and whether the respondent was living with family. The latter factor was assessed because of possible differences in behavior owing to concerns about infection at home. We also obtained information about health literacy associated with engaging in health behaviors. For health literacy, we used the five-item version of the validated scale, Communicative and Critical Health Literacy, developed by Ishikawa et al. [29].

#### 2.3.1. COVID-19 Vaccination Behavior

We asked whether participants had received a first COVID-19 vaccination, with response options of either yes or no. Because the survey was conducted five months after the start of vaccination of the general public, only the first vaccination was included in the survey.

#### 2.3.2. Trust in COVID-19 Information Sources

We assessed participants’ trust in eight different media sources of information: the government, prefectural governors, experts, celebrities, physicians, infected patients, bloggers, and social media. We also assessed trust in three interpersonal sources of information: primary care physicians, family, and friends. In Japan, with the declaration of the COVID-19 pandemic, the Novel Coronavirus Expert Committee was organized, comprising experts in infectious disease control. Experts in this study were defined as members of this organization. Participants were asked how much they trusted COVID-19 vaccine information from each of the above sources, using a single question (e.g., “How much do you trust [the government] as a source of COVID-19 vaccine information?”). Each item was assessed on a 10-point scale, ranging from 1 (do not trust at all) to 10 (trust very much). This measure was adapted from a previous study [30,31,32,33].

### 2.4. Ethical Considerations

Our study protocol was approved by the ethical review committee at the Graduate School of Medicine, The University of Tokyo (number 11270). This study was carried out in accordance with the Declaration of Helsinki. All participants gave their written informed consent.

### 2.5. Statistical Analysis

Descriptive statistics were calculated for each sociodemographic variable. The continuous variables age and health literacy were categorized, with age divided into six age groups in 10-year increments and health literacy divided into two categories based on the median age. We then performed a crude analysis of the association of each sociodemographic variable and trust in information sources with the first COVID-19 vaccination status using the χ-squared test. Multiple logistic regression analysis was performed to examine the associations among COVID-19 vaccination status, sociodemographic variables, and trust in information sources. In Model 1, sociodemographic variables (sex, age, education, household income (USD), employment, place of residence, presence of underlying medical conditions, living with family, and health literacy) were included as explanatory variables. In Model 2, in addition to the variables in Model 1, we included trust in the eight media sources of information. In Model 3, in addition to the variables in Model 2, we included trust in the three interpersonal information sources. The objective variable was COVID-19 vaccination status (vaccinated = 1, unvaccinated = 0) in all models. There were no missing values owing to the specifications of the online survey. All tests were two-sided, and the significance level was set at 5%. We used IBM SPSS version 25 (IBM Corp., Armonk, NY, USA) for the analysis.

## 3. Results

Table 1 presents the participants’ characteristics. As stated in Section 2.2, the distribution of participants’ sex, age, and prefecture of residence was matched with that of Japan’s general population. As for educational background, 56.2% of participants had a university degree or higher. Approximately half of the participants had annual household incomes under USD 15,000, and approximately half were employed full-time.

Table 2 shows the results of multiple logistic regression analysis with COVID-19 vaccination status as the outcome variable and sociodemographic characteristics as explanatory variables (Model 1). By sex, women were vaccinated significantly more frequently than men. Older people were more likely to be vaccinated, especially those over the age of 50 years, which was significant. For household income, respondents who earned more than USD 15,000 were vaccinated significantly more frequently than those with household incomes less than USD 15,000. Regarding the place of residence, participants living in Chiba Prefecture were significantly more vaccinated than those living in other prefectures. Regarding underlying disease, participants with an underlying disease were significantly more vaccinated than those without an underlying disease. Participants who lived with their families were significantly more vaccinated than those who did not.

Table 3 shows the results of multiple logistic regression analysis with COVID-19 vaccination status as the outcome variable and sociodemographic characteristics and trust in information sources as explanatory variables. In Model 2, trust in experts as a media source of information showed a significant positive association with vaccination behavior (adjusted odds ratio (OR) 1.157; 95% confidence interval (CI), 1.017–1.31). In Model 3, trust in primary care physicians as an interpersonal source of information had a significant positive association with vaccination behavior (adjusted OR 1.076; 95% CI, 1.006–1.150). When we included trust in information sources in the model, sex, the presence of underlying medical conditions, and living with the family were no longer significantly associated with vaccination behavior.

## 4. Discussion

This study examined the association between trust in 11 different sources of information and COVID-19 vaccination behavior during the introduction of COVID-19 vaccines under emergency approval in Japan. Previous studies on vaccination behavior have focused on cognitive aspects such as vaccine acceptance and behavioral intentions [8,9,10,11,12,13]. This study adds to those findings in terms of our results on actual vaccination behavior during the phase of introducing COVID-19 vaccines under emergency approval. Vaccination under emergency approval requires acceptance of a greater level of uncertainty than that associated with approved vaccines. The results of this study will be useful for vaccine communication in the current COVID-19 pandemic as well as in future outbreaks of emerging infectious diseases.

Regarding sociodemographic characteristics, in line with studies on COVID-19 vaccination intention conducted in the United Kingdom and the U.S., age and household income were associated with vaccination behavior [34,35,36]. Given that COVID-19 is more likely to cause severe disease in older adults, vaccination of older people began earlier than vaccination of younger adults in Japan [37]. The perception of a higher risk of severe disease and a priority vaccination program may have prompted vaccination among older people. Younger people place a lower priority on vaccination compared with commitments they have regarding their job or friends [38]. Therefore, accessibility is important to vaccination behavior [38]. Targeted vaccine communication should be provided for low-income households and younger people, such as providing mass vaccination programs for younger people at universities and workplaces.

Regarding trust in media sources of information, contrary to previous studies, we found that trust in the government was not associated with vaccination behavior [13,30,39,40,41,42,43,44,45,46,47,48]. This result may be interpreted as follows. First, after the COVID-19 pandemic was declared, Japan’s government disseminated strongly persuasive daily messages encouraging compliance with preventive behaviors. Recipients’ trust decreases when they perceive that the message source intends to persuade [49]. This is because the source’s motivation for persuasion is perceived by recipients as including personal benefit for the source itself [50]. At the same time, when people perceive that their freedom may be restricted, psychological reactance is generated [51]. Messages from the government seeking to compel people to engage in certain behaviors (e.g., wearing a mask) or that dictate changes in daily behavior (e.g., improving ventilation or staying at home) may increase psychological reactance. Second, cooperation based on people’s trust in information sources is influenced by their perception of the source’s motivation (Does the source have my best interests at heart?) and competence (Has the source been competent and trustworthy in the past?) [4]. In March 2021, the Japanese government decided to open the Tokyo Olympics in July 2021 at the same time as vaccination began. With the opening of the Olympics, the number of infected cases reached a record high in Japan. The public’s perception of a change in the government’s motivation for infection control may have reduced their trust in and cooperation with the Japanese government.

In our study, trust in public health experts as a media source of information was positively associated with vaccination behavior. Trust in sources reflects the perception that the source is competent and has expertise related to the topic [52]. It has also been reported that experts perceived to have expertise are more persuasive when persuading individuals to adopt health behaviors [53,54,55]. It is possible that the public’s perception of having expertise about the COVID-19 vaccine made the expert’s message more persuasive. Previous studies have reported that higher levels of trust in government medical expert organizations, such as the CDC, have a positive impact on vaccination [56,57]. The CDC’s scientific advice is not always reflected in policy. However, a certain distance from politics ensures scientific neutrality. Maintaining this relationship may help to minimize public distrust of one information source or the other. As mentioned earlier, the Japanese government made a political decision to host the Olympic Games that may have affected public trust in the timing of COVID-19 vaccination. Government and expert organizations are the two pillars of infection control. Even if trust in the government declines, if trust in expert organizations is maintained, it may be possible to promote vaccination by disseminating messages from that source. There is no public health expert organization like the U.S. CDC in Japan. However, the results of our study showed that the expert meetings that were temporarily organized during the COVID-19 pandemic in Japan gained a certain level of public trust and that trust was associated with vaccination behavior. Neighboring China and South Korea have established a CDC, which is responsible for public health activities, including infectious disease control [58,59]. This pandemic presents an opportunity for Japan to establish a similar expert organization that can continuously address public health issues and effectively communicate with the public.

Regarding interpersonal sources of information, greater trust in the primary physician as a source of information was positively associated with vaccination behavior. Previous studies have reported that physician recommendations are a strong promoter of vaccination and that doctor-patient communication is associated with patients’ vaccination attitudes and behaviors [27,60,61,62,63,64]. The results of this and previous studies indicate that primary care physicians are a persuasive and important resource in health communication when promoting emergency vaccination. One reason for the high level of trust in physicians among the Japanese is that Japan has a long history of paternalistic medicine; even today, there is a high regard for what physicians say [65]. How a health care system has performed in the past, and the perceived values of that system play an important role in the trust-building process [4]. Primary care physicians may be important in promoting vaccination among the Japanese. Concern about side effects is a strong barrier to vaccination [36,66,67]. High-quality doctor-patient communication, such as ensuring that medical decisions are based on the patient’s needs and values, helps patients build knowledge about their treatment and relieve their concerns about side effects [68]. There is no general practitioner system in Japan. The results of the present study provide a rationale for recommending that people develop a relationship with a trusted primary care physician to prepare for future public health challenges.

Regarding physicians in the media, trust in this information source was not associated with vaccination, despite these physicians having the same medical training and license as primary care physicians. This result can be interpreted as follows. First, nonverbal immediacy involves nonverbal communication that reinforces perceptions of intimacy, such as approaching behavior and positive nods; nonverbal immediacy has been reported to have positive effects on persuasion [69]. However, the use of nonverbal communication to develop psychological proximity and intimacy with an audience is limited via the media. This may account for greater trust in primary care physicians who can communicate face-to-face. Second, physicians usually communicate with patients to reach an agreement with them, not to persuade them. Physicians in the media, who provide information to much wider audiences than in face-to-face communication with patients, may have experienced a moral dilemma about whether to attempt to persuade people who may have anxiety and concerns about the rapidly developed COVID-19 vaccines [70]. Therefore, these physicians may not have been able to make definitive recommendations via the media regarding COVID-19 vaccines. Consequently, an association between trust in physicians in the media and vaccination behavior may not have been detected.

An important point to consider is that the media’s reporting style may affect the public’s trust in experts and physicians [71,72]. In the case of the human papillomavirus vaccine in Japan, trust in physicians and researchers was lost owing to the manner in which the media reported adverse events [73]. Public health experts and physicians should actively work with the media to avoid spreading misinformation and to maintain the trust of the public. Toward this aim, a previous study recommended that experts should be easily accessible to journalists, specialist medical reporters should be trained, and reliable and useful information resources [74].

The results of this study should be interpreted in light of several limitations. First, because this was a cross-sectional study, our results cannot be used to determine causality. Therefore, in our discussion, we relied on previous studies and existing theories to support our causal inferences. Second, participants were registered members of a survey panel of an Internet company. The possibility of clicking on a random answer cannot be ruled out. However, we included a quality check question, which respondents were required to read carefully. We excluded participants who provided unreliable responses. There were no significant differences in sociodemographic characteristics between the analyzed and excluded participants. Third, the measures of trust in information sources used in this study were single items and were not validated. We adopted items frequently used in previous studies so that our results would be comparable to results obtained during other infectious disease pandemics and results from other countries. Fourth, Participants younger than 65 years had a month and a half shorter time from the start of vaccination to the survey than participants older than 65 years. At the time of the survey, the vaccination rate among the Japanese population was about 88% for those aged 65 and older and about 58% for those under 65 (however, it was about 72% for those aged 50 and older). The results of the multiple regression analysis, although adjusted for age, may have been influenced by differences in vaccination duration. Finally, this study focused on information sources but not on information content. Future studies should address both the source and content of information and examine the type of content disseminated by each information source that gains people’s trust and influences their behavior.

Despite the above limitations, this study was the first to identify associations with trust in sources of information related to COVID-19 vaccination under emergency use authorization in Japan. These findings have important implications for public health communication during a pandemic.

## 5. Conclusions

Greater trust in experts as a source of media information and primary care physicians as a source of interpersonal information were associated with vaccination with the first dose of the COVID-19 vaccine, approved under emergency use authorization in Japan. Health communication strategies to promote vaccination behavior that require people to accept high levels of uncertainty should focus on experts and primary care physicians as persuasive information sources.

## Figures and Tables

**Table 1 vaccines-11-00233-t001:** Characteristics of study participants (*n* = 784).

	Total (*n* = 784)		Vaccinated (*n* = 546)		Unvaccinated (*n* = 238)		
Variables	*n*	%	*n*	%	*n*	%	*p* ^‡^
Sex							
Male	396	50.5	273	50.0	123	51.7	0.665
Female	388	49.5	273	50.0	115	48.3	
Age							<0.001 *
<30	94	12.0	53	9.7	41	17.2	
30–39	143	18.2	89	16.3	54	22.7	
40–49	197	25.1	123	22.5	74	31.1	
50–59	182	23.2	134	24.5	48	20.2	
Over 60	168	21.4	147	26.9	21	8.8	
Education							0.007 *
Junior high school	12	1.5	6	1.1	6	2.5	
High school	156	19.9	96	17.6	60	25.2	
Vocational school/Junior college	175	22.3	116	21.2	59	24.8	
University	379	48.3	278	50.9	101	42.4	
Graduate school	62	7.9	50	9.2	12	5.0	
Household income (USD)							0.001 *
>15,000	334	42.6	31	5.7	33	13.9	
15,000–43,500	66	8.4	222	40.7	98	41.2	
>43,500	64	8.2	47	8.6	19	8.0	
Unknown	320	40.8	246	45.1	88	37.0	
Employment status							0.569
Employed full-time	415	52.9	292	53.5	123	51.7	
Employed part-time	179	22.8	119	21.8	60	25.2	
Retired/Unemployed	190	24.2	135	24.7	55	23.1	
Place of residence							0.069
Hokkaido	73	9.3	46	8.4	27	11.3	
Kyoto	37	4.7	25	4.6	12	5.0	
Hyougo	66	8.4	39	7.1	27	11.3	
Chiba	75	9.6	52	9.5	23	9.7	
Saitama	100	12.8	71	13.0	71	13.0	
Kanagawa	123	15.7	87	15.9	36	15.1	
Osaka	115	14.7	74	13.6	41	17.2	
Tokyo	195	24.9	152	27.8	43	18.1	
Underlying medical conditions							<0.001 *
Yes	635	81.0	123	22.5	26	10.9	
No	149	19.0	423	77.5	212	89.1	
Living with family							0.002 *
Yes	613	78.2	443	81.1	170	71.4	
No	171	21.8	103	18.9	68	28.6	
Health literacy ^†^							0.017 *
Low (4>)	355	45.3	232	42.5	123	51.7	
High (≥4)	429	54.7	314	57.5	115	48.3	

SD, standard deviation. * *p* < 0.05; † Health literacy was measured using the Communicative and Critical Health Literacy instrument, ‡ Chi-square test between vaccinated and unvaccinated groups.

**Table 2 vaccines-11-00233-t002:** Association between sociodemographic characteristics and COVID-19 vaccination (*n* = 784).

Model 1				
Variables	OR	95% CI		*p*
Sex				
Male	Ref			
Female	1.467	1.006	2.141	0.047 *
Age				
<30	Ref			
30–39	1.182	0.673	2.077	0.560
40–49	1.219	0.706	2.106	0.478
50–59	2.438	1.368	4.346	0.003 *
Over 60	6.331	3.169	12.648	<0.001 *
Education				
Junior high school	Ref			
High school	1.336	0.387	4.617	0.647
Vocational school/Junior college	1.464	0.420	5.102	0.549
University	2.106	0.618	7.182	0.234
Graduate school	3.596	0.913	14.162	0.067
Household income (USD)				
<15,000	Ref			
15,000–43,500	2.112	1.116	3.997	0.022 *
>43,500	2.780	1.223	6.319	0.015 *
Unknown	2.285	1.158	4.508	0.017 *
Employment status				
Employed full-time	Ref			
Employed part-time	0.773	0.488	1.226	0.274
Retired/Unemployed	0.620	0.377	1.018	0.059
Place of residence				
Hokkaido	Ref			
Kyoto	1.204	0.593	2.445	0.607
Hyougo	1.329	0.629	2.808	0.456
Chiba	2.050	1.076	3.906	0.029 *
Saitama	1.385	0.701	2.734	0.348
Kanagawa	1.249	0.509	3.066	0.627
Osaka	1.017	0.522	1.980	0.960
Tokyo	0.691	0.325	1.467	0.336
Underlying medical conditions				
No	Ref			
Yes	1.885	1.136	3.126	0.014 *
Living with family				
No	Ref			
Yes	1.611	1.060	2.450	0.026 *
Health literacy				
Low	Ref			
High	1.246	0.890	1.744	0.201

OR, odds ratio; CI, confidence interval; Ref, reference. * *p* < 0.05; Odds ratios were calculated using multivariate logistic regression analysis.

**Table 3 vaccines-11-00233-t003:** Association between trust in each source and COVID-19 vaccination (*n* = 784).

	Model 2				Model 3			
Variables	OR	95% CI		*p*	OR	95% CI		*p*
Media Sources								
Government	0.963	0.847	1.096	0.567	0.960	0.843	1.094	0.538
Prefectural governors	1.099	0.952	1.268	0.196	1.091	0.944	1.262	0.239
Experts	1.157	1.017	1.317	0.026	1.167	1.024	1.331	0.021 *
Celebrities	0.922	0.833	1.021	0.118	0.914	0.825	1.013	0.085
Physicians	1.045	0.923	1.183	0.490	1.010	0.888	1.149	0.874
Infected patients	0.974	0.876	1.082	0.622	0.959	0.859	1.069	0.448
Social media	1.054	0.962	1.155	0.258	1.062	0.969	1.165	0.200
Bloggers	1.001	0.914	1.096	0.985	0.972	0.884	1.067	0.548
Interpersonal Sources								
Friends					0.996	0.915	1.083	0.920
Family					1.069	0.983	1.163	0.119
Primary care physician					1.076	1.006	1.150	0.033 *

OR, odds ratio; CI, confidence interval; Ref, reference. * *p* < 0.05; Odds ratios were adjusted for sex, age, education, Household income, Employment status, Place of residence, Underlying medical conditions, Living with family, and Health literacy by multivariate logistic regression analysis.

## Data Availability

The datasets generated and/or analyzed during the current study are available from the corresponding author on reasonable request.

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
