# Peer review of "Associations between Vaccination Behavior and Trust in Information Sources Regarding COVID-19 Vaccines under Emergency Approval in Japan: A Cross-Sectional Study"

_vaccines, 2023, doi:10.3390/vaccines11020233_

Round 1

Reviewer 1 Report

- Lines 89-92: If the following is stated `For the general public, vaccination of people 65 years and older started in April 2021 and vaccination of people younger than 65 years started in June 2021.`, explain how this study (conducted during August 2021) was done as it is currently written, i.e. `Surveys were conducted at 5 months after the start of vaccination of the general public.`. Namely, `general population` is not made up of only those older than 65 years, but also those younger than 65 years. Explain and correct, because this period (`5 months`) can significantly influence the results in this study.  

- Line 93: State whether the vaccination against COVID-19 in Japan was mandatory (and for which indications), or recommended (and for which indications), or not.      - Line 171: Add the other variables where statistical significance was reached in Model 1 (Place of residence - Chiba, Underlying medical conditions, Living with family).    - Line 304: Discuss, as a potential limitation of this study, the influence of different periods (5 months versus 2 months) until the realization of study (August 2021), looking from the beginning of vaccination against COVID-19 which among people 65 years and older started in April 2021 and vaccination of people younger than 65 years started in June 2021.   

Author Response

Response to Reviewer 1:

- Lines 89-92: If the following is stated `For the general public, vaccination of people 65 years and older started in April 2021 and vaccination of people younger than 65 years started in June 2021.`, explain how this study (conducted during August 2021) was done as it is currently written, i.e. `Surveys were conducted at 5 months after the start of vaccination of the general public.`. Namely, `general population` is not made up of only those older than 65 years, but also those younger than 65 years. Explain and correct, because this period (`5 months`) can significantly influence the results in this study.  
Reply: We have modified the introduction section as follows.
(Line 91-) For the general public, vaccination of people 65 years and older started on April 12, 2021, and vaccination of people younger than 65 years started in June 1, 2021. Surveys were conducted 5 months after the start of vaccination for the general public aged 65 years and older and 2.5 months after the start of vaccination for the general public aged less than 65 years.

- Line 93: State whether the vaccination against COVID-19 in Japan was mandatory (and for which indications), or recommended (and for which indications), or not.     
Reply: We have added the following to the study design section regarding the status of COVID-19 vaccine recommendations in Japan.
(Line 97) At the time of the survey, the legal status of the COVID-19 vaccine in Japan was "Duty to Endeavor" for people aged 12 and older. This means that vaccination was not mandatory, but efforts had to be made to vaccinate.

- Line 171: Add the other variables where statistical significance was reached in Model 1 (Place of residence - Chiba, Underlying medical conditions, Living with family).   
Reply: We have added the following to the Results section.
(Line 177-) Regarding place of residence, participants living in Chiba Prefecture were significantly more vaccinated than those living in other prefectures. Regarding underlying disease, participants with an underlying disease were significantly more vaccinated than those without an underlying disease. Participants who lived with their families were significantly more vaccinated than those who did not.

- Line 304: Discuss, as a potential limitation of this study, the influence of different periods (5 months versus 2 months) until the realization of study (August 2021), looking from the beginning of vaccination against COVID-19 which among people 65 years and older started in April 2021 and vaccination of people younger than 65 years started in June 2021.   

Reply: We have added the following to the limitation section of the Discussion.

Participants younger than 65 years had a month and a half shorter time from the start of vaccination to the survey than participants older than 65 years. At the time of the survey, the vaccination rate among the Japanese population was about 88% for those aged 65 and older and about 58% for those under 65 (However, about 72% for those aged 50 and older). The results of the multiple regression analysis, although adjusted for age, may have been influenced by differences in vaccination duration.

Reviewer 2 Report

I have read the manuscript with interest. The authors have studied vaccination behaviour for COVID-19 first dose against sources of information. The introduction is particularly well laid out with the knowledge gap and rationale for study question and study design clearly laid out. The study method is adequately described and the manuscript is well-written. I have only minor suggestions for improvement. 

Minor suggestions 

1. Introduction - Line 30 - "with effective treatment for COVID-19 infection lacking,..." this phrase is misleading as we now have effective treatments for COVID-19 available. Please revise. 
2. Line 32 - "...the highly effective COVID-19 vaccine.." please specify by name and manufacturer which vaccine is referred to here. 

3. Line 67 - "Trust in ..... associated with vaccination." This statement needs a citation. 

4, Methods - Please provide further detail on sampling, specifically - how many participants were invited, what was the response rate and how many responses were excluded and for what reasons ? 

5. Tables - Please note that highlighting or otherwise indicating the significant variables is desirable to improve readability of tables 

6. Table 1 and 2 , Variable household income - I believe that the first group should be <15,000. If that is so please correct the error. 

7. Discussion, Line 190 - "Previous studies on vaccination ...." the statement needs citation. 

8. Discussion could be improved with some discussion of vaccination behavior for other vaccines in relation to sources of information and through clearly articulating the theoretical framework lens through which authors are interpreting their findings. 

Author Response

Response to Reviewer 2:

  1. Introduction - Line 30 - "with effective treatment for COVID-19 infection lacking,..." this phrase is misleading as we now have effective treatments for COVID-19 available. Please revise. 
    Reply: We apologize for the misleading text we included. We have removed the relevant text.

  2. Line 32 - "...the highly effective COVID-19 vaccine.." please specify by name and manufacturer which vaccine is referred to here. 
    Reply: We have added the following to the introductory section.
    (Line 31-) COVID-19 vaccines were developed rapidly, and by the end of 2020, the BNT162b2 (Pfizer–BioNTech) vaccine, mRNA-1273 (Moderna) vaccine, and Ad26.COV2.S (Johnson & Johnson–Janssen) vaccine had received Emergency Use Authorization (EUA) from the U.S. Food and Drug Administration [1,2].
  3. Line 67 - "Trust in ..... associated with vaccination." This statement needs a citation. 
    Reply: We have added the following citations
    Trust in information sources such as information received in face-to-face interpersonal communication has also been reported to be associated with vaccination [18,19,21-28].

4, Methods - Please provide further detail on sampling, specifically - how many participants were invited, what was the response rate and how many responses were excluded and for what reasons ? 
(Line 108-) An email was sent to 8,774 survey company enrollees who met the eligibility criteria, excluding health care providers. Of those, 1565 e-mail recipients accessed the text detailing the survey and indicated their willingness to participate. Of those, age, gender, and prefecture of residence were sampled to match Japan's general population, and 788 were invited to participate in the web survey. Four respondents who reported that they were unable to be vaccinated due to physical conditions were excluded. 784 were included in the final analysis.

  1. Tables - Please note that highlighting or otherwise indicating the significant variables is desirable to improve readability of tables 
    Reply: We have marked * for p-values less than 0.05 in the tables.

  1. Table 1 and 2 , Variable household income - I believe that the first group should be <15,000. If that is so please correct the error. 
    Reply: We apologize for the typographical error. We have modified Tables.

  1. Discussion, Line 190 - "Previous studies on vaccination ...." the statement needs citation. 

Reply: We have added the following citations.

(Line 202-) Previous studies on vaccination behavior have focused on cognitive aspects such as vaccine acceptance and behavioral intentions [8-13].

  1. Discussion could be improved with some discussion of vaccination behavior for other vaccines in relation to sources of information and through clearly articulating the theoretical framework lens through which authors are interpreting their findings. 
    Reply: Thank you for your advice. The discussion in this paper is based primarily on persuasion theory. We have added persuasion theory to our interpretation of the "expert" results as follows.

(Line 242-) Trust in sources reflects the perception that the source is competent and has expertise related to the topic [53]. It has also been reported that experts perceived to have expertise are more persuasive when persuading individuals to adopt health behaviors [54-56]. It is possible that the public's perception of having expertise about the COVID-19 vaccine made the expert's message more persuasive.